structural engineering

graphic statics, structural stability, Maxwell reciprocal diagrams, prestress

**Author for correspondence:**
Allan McRobie
e-mail: fam20@cam.ac.uk

# Stability of trusses by graphic statics

Allan McRobie[1], Cameron Millar[1] and William F. Baker[2]

[1]Cambridge University Engineering Department, Trumpington St, Cambridge CB2 1PZ, UK
[2]Skidmore Owings and Merrill, Chicago, IL, USA

AM, 0000-0002-6610-5927

This paper presents a graphical method for determining the linearized stiffness and stability of prestressed trusses consisting of rigid bars connected at pinned joints and which possess kinematic freedoms. Key to the construction are the rectangular areas which combine the reciprocal form and force diagrams in the unified Maxwell–Minkowski diagram. The area of each such rectangle is the product of the bar tension and the bar length, and this corresponds to the rotational stiffness of the bar that arises due to the axial force that it carries. The prestress stability of any kinematic freedom may then be assessed using a weighted sum of these areas. The method is generalized to describe the out-of-plane stability of two-dimensional trusses, and to describe three-dimensional trusses in general. The paper also gives a graphical representation of the 'product forces' that were introduced by Pellegrino and Calladine to describe the prestress stability of trusses.

## 1. Introduction

Graphic statics has traditionally concerned itself with the equilibrium of pin-jointed trusses. Here, we extend the methods to describe the stability of such structures. The case of interest is known as 'prestress stability' and it concerns structures which possess at least one mechanism which may be stabilized or destabilized by the geometric stiffness effects that arise from the axial forces carried by the structural members.

Following Maxwell [1,2] the graphic analysis of the equilibrium of a pin-jointed truss is accomplished via a pair of reciprocal diagrams, with the truss geometry given by the *form diagram*, while the *force diagram* represents an equilibrium set of axial forces within that truss. In this paper, the methodology is extended to quantify the stiffness that arises due to the presence of forces within the members, allowing the stability to be assessed. Figure 1 shows a simple example where a pair of hinged bars has a mechanism involving lateral deflection. An axial compressive force would cause the system to buckle, while a tension force would tend to pull the structure straight, adding stiffness such that the structure would even be capable of carrying some transverse load.

**Figure 1.** A bar with a central hinge pulls straight under tension but is unstable in compression and buckles laterally.

Such tension stiffening and compression de-stiffening are not described by the simplest structural theories where equilibrium equations are written at the undeformed configuration. While better theories can describe buckling and tension stiffening, the inability of traditional matrix formulations of structural mechanics to undergo such destabilization persists even into the large finite-element packages routinely used in structural analysis. An analyst will need to create a considerably more sophisticated model in order to successfully simulate such stabilizing and destabilizing effects. While it would be tempting to describe these as higher-order effects, this is not strictly correct: the instability phenomena have been missed because the equilibrium equations were written at the *undeformed* shape.

In this paper, we thus require a clear statement of structural behaviour which considers equilibrium at the deformed shape. Building on work of Pellegrino & Calladine [3–6] and Connelly & Whiteley [7], Guest [8] provides a matrix formulation of the exact tangent stiffness of any prestressed pin-jointed truss with respect to perturbations about its deformed equilibrium configuration. This matrix decomposes into two components similar to the possibly more familiar decomposition of a stiffness matrix into a material part and a geometric part, with the latter representing the effects of forces on the overall system stiffness. In the Guest description, it is the closely related *stress matrix* that plays this role, and that will be the primary study of this paper. This paper begins by showing how the terms of the stress matrix may be represented first vectorially and then graphically, giving examples of simple structures.

Before looking at stability, graphic statics is first employed in the initial equilibrium step, establishing the reciprocal diagrams of form and force. This may be readily accomplished by the traditional technique of constructing lines in the plane. However, the analysis gains power by the introduction of polyhedral Airy stress functions, whose existence follows from Maxwell's observation [1] that reciprocal form and force diagrams are the two-dimensional projections of dual polyhedra.

Once an equilibrium state is established, the second use of graphic statics here is to identify mechanisms. For in-plane mechanisms of two-dimensional trusses, this follows the procedure described in McRobie *et al.* [9] whereby an increment to the dual Airy stress function over the force diagram leads to increments in the coordinates of the form diagram. These small displacements give bar increments which are necessarily parallel to the original bar directions and thus when the nodal displacement vectors are rotated by 90°, they necessarily correspond to an infinitesimal inextensional mechanism of the original structure. Figure 4 later gives an example of this method. Geometrical methods for detecting mechanisms in three-dimensional trusses are described in McRobie *et al.* [10], but these need not concern us here.

The other graphical element that is key to the construction is the Maxwell–Minkowski diagram, which combines the form and the force diagrams into a single geometric object [11,12]. In such a diagram, the polygons that comprise the form and the force diagrams are connected by a set of intervening rectangles of dimensions bar length $L$ by tension force $T$. The area of the rectangle thus gives geometric manifestation to the internal work $TL$ associated with that bar's contribution to the summation in the Maxwell Load Path Theorem [2,12]. Similarly, the aspect ratios of the rectangles give geometric representation to the *tension coefficients* (also called the *force densities*) $T/L$ that are used in a variety of approaches to structural analysis (e.g. [13]). The novel interpretation of this paper is that the rectangular areas $TL$ represent the rotational stiffness of the stressed bars, and a weighted sum of these stiffnesses determines the stability.

Throughout this paper, the emphasis on the graphical presentation of problems usually addressed by matrix-based techniques is in line with many intentions that have led to the current renaissance of graphic statics generally. That is, the motivation is not driven by any particular desire for solution procedures that are more efficient or robust, given the phenomenal advances in computational speed and algorithm reliability over the last half century. Rather, the graphical procedures provide an alternative ansatz wherein a designer can gain a more intuitive understanding of the mechanics of structural behaviour. In the examples that follow, it is hoped that the reader will begin to 'see' the basis of stability in any particular problem, in a way that might be difficult to envisage when using a matrix algorithm.

Moreover, the duality of form and force that underlies graphic statics is particularly suited to early-stage structural design, where a structural form may not yet be known. In that setting, graphic statics can

provide a powerful environment in which to optimize structural efficiency [14]. The newly found ability to visualize forces, load paths and now bar stiffnesses is all part of an approach aimed at promoting greater understanding.

This paper relates to others that the authors are currently in the process of publishing on this subject. It expands the initial ideas presented in the conference paper of Millar *et al.* [15]. A detailed case study applying this approach to the stability of Robert Maillart's historically significant roof design at Chiasso, Switzerland is presented in McRobie *et al.* [16]. The large deformation case is considered in Millar *et al.* [17], and a paper is in preparation for the case when the axial rigidity constraint is relaxed [18].

We begin by stating the matrix formulation of geometric stability. This is the linearized version of structural analysis that uses small displacement theory, but writes the equilibrium equation at the deformed configuration. There are a number of different descriptions of this, and we focus here on two of them; the 1980s theory of Pellegrino & Calladine [4,19] with its use of so-called 'product forces', and the later formulation by Guest [8] that has its roots in mathematical rigidity theory, particularly that of Connelly & Whiteley [7]. Guest's version is presented first, even though that does not match the historical chronology.

# 2. Guest's rigidity theory formulation

Guest [8] presents the derivation from first principles that shows that the (linearized) change in total potential energy when a pin-jointed truss undergoes a small displacement from an equilibrium configuration is

$$W = \frac{1}{2}\mathbf{d}^T\mathbf{K}\mathbf{d},$$ (2.1)

where $\mathbf{d}$ is the vector of nodal displacements from the equilibrium configuration and $\mathbf{K}$ is the *total tangent stiffness matrix*. This is composed of two terms

$$\mathbf{K} = \mathbf{A}\hat{\mathbf{G}}\mathbf{A}^T + \mathbf{S},$$ (2.2)

where the first term involves the equilibrium matrix $\mathbf{A}$ and a diagonal matrix $\hat{\mathbf{G}}$ of modified axial stiffnesses. The second term $\mathbf{S}$ is the so-called *stress matrix*. The first term depends on the material elastic modulus. However, in this paper, bars are assumed to be axially rigid and we only investigate mechanisms that exist in the left null space of $\mathbf{A}$ such that it will be the stress matrix $\mathbf{S}$ that will be central to the analysis.

It should be noted that the 'stress matrix' terminology arises from rigidity theory, and does not accord with standard engineering usage. For rigidity theorists, the word 'stress' refers to the quantity $T/L$, which is called the 'tension coefficient' or 'force density' in engineering.

Guest's derivation gives the stress matrix $\mathbf{S}$ as

$$\mathbf{S} = \begin{bmatrix} \mathbf{s}_{11} & \mathbf{s}_{12} & \cdots & \mathbf{s}_{1n} \\ \mathbf{s}_{21} & \mathbf{s}_{22} & & \\ \vdots & & \ddots & \\ \mathbf{s}_{n1} & & & \mathbf{s}_{nn} \end{bmatrix},$$ (2.3)

where, depending on the dimensionality of the problem, the $2 \times 2$ or $3 \times 3$ submatrices $\mathbf{s}_{ij}$ are given by

$$\mathbf{s}_{ii} = \hat{t}_{ii}\mathbf{I} \quad \text{and} \quad \mathbf{s}_{ij} = -\hat{t}_{ij}\mathbf{I},$$ (2.4)

with $\hat{t}_{ii}$ being the sum of the tension coefficients $T/L$ of all bars that meet at node $i$, and $\hat{t}_{ij}$ being the tension coefficient of the bar joining nodes $i$ and $j$ (and is zero if there is no such bar).

Subsequent papers will present results for extensional bars and multiple mechanisms [16–18], but this paper restricts attention to the case where all bars are rigid and there is a single infinitesimal mechanism $\mathbf{d} = \mathbf{m}$ which involves no bar extensions. Since $\mathbf{A}^T\mathbf{m} = \mathbf{0}$, it follows that the only work done is that part associated with the stress matrix. When the structure is articulated into the shape $\lambda\mathbf{m}$, the work done is

$$W = \frac{1}{2}\,\mathbf{m}^T\mathbf{S}\mathbf{m}\,\lambda^2 = \frac{1}{2}K_s\lambda^2.$$ (2.5)

The stability of the structure against articulation in that mechanism is then given by the sign of the generalized stiffness $K_s = \mathbf{m}^T\mathbf{S}\mathbf{m}$. This is a scalar quantity and the system of internal bar forces stabilizes the mechanism if $K_s$ is positive.

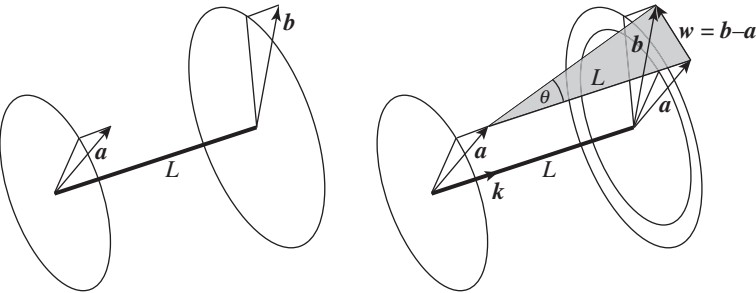

**Figure 2.** Bar displacement vectors.

## 2.1. Simplification of Guest's formulation

For trusses consisting of inextensional bars, Guest's matrix formulation may be rewritten as a simple sum. Consider an inextensional bar from node $i$ to node $j$ with tension coefficient $\hat{t}$. Let the nodal displacements be $\mathbf{a}$ at node $i$ and $\mathbf{b}$ at node $j$, as per figure 2. From equations (2.4) and (2.5), the contribution of this bar to the work $W$ is

$$\frac{1}{2}\hat{t}\begin{bmatrix} \mathbf{a}^T, & \mathbf{b}^T \end{bmatrix}\begin{bmatrix} \mathbf{I} & -\mathbf{I} \\ -\mathbf{I} & \mathbf{I} \end{bmatrix}\begin{bmatrix} \mathbf{a} \\ \mathbf{b} \end{bmatrix} = \frac{1}{2}\hat{t}(\mathbf{b}-\mathbf{a})\cdot(\mathbf{b}-\mathbf{a}). \tag{2.6}$$

The total work is thus

$$W = \frac{1}{2}\sum_{\text{bars}}\hat{t}(\mathbf{b}-\mathbf{a})\cdot(\mathbf{b}-\mathbf{a}). \tag{2.7}$$

Displacement vectors $\mathbf{a}$ and $\mathbf{b}$ may be decomposed into components parallel and perpendicular to the bar. Subtracting $\mathbf{a}$ from $\mathbf{b}$ removes the parallel components, leaving only the perpendicular components. Writing the magnitude of the differential lateral displacement as $w = |\mathbf{b}-\mathbf{a}|$ the contribution of this bar to the total potential energy is

$$W = \frac{1}{2}\hat{t}w^2 = T\frac{w^2}{2L}. \tag{2.8}$$

The $w^2/2L$ term is familiar from traditional approaches to stability as the axial end-shortening $y = L - L\cos\theta = L - L(1-(1/2)(w/L)^2 + \cdots) \approx w^2/2L$ that occurs due to a differential lateral displacement $w$ between the bar ends. Indeed, this formula for the rigid bar case follows immediately from the total potential energy. In the deformed configuration, the contribution of each bar to the total potential energy $\Pi$ is

$$\Pi = \int_0^L \text{strain energy density } \mathrm{d}l - Py, \tag{2.9}$$

where $-Py$ is the work done by the loads applied at the bar ends. Here, $P$ and $y$ are work conjugate, and since $y$ is an end-shortening, then $P$ is the compressive axial force. For inextensible bars, the strain energy terms are zero, leaving

$$W = \sum_{\text{bars}} -Py = \sum_{\text{bars}} Ty = \sum_{\text{bars}} T\frac{w^2}{2L} \tag{2.10}$$

as earlier. This makes clear that the external work done by the applied forces is accounted for in the energy summation over the bars, and does not need to be included separately.

Letting $\mathbf{k}$ be the unit vector along the bar, then during the small deformation, the bar rotates by $(\mathbf{k}\times(\mathbf{b}-\mathbf{a}))/L$ about an axis perpendicular to the bar. This rotation has magnitude $\theta = w/L$, and the total potential energy associated with the deformation of the whole structure is thus

$$W = \frac{1}{2}\sum_{\text{bars}}(TL)\theta^2. \tag{2.11}$$

Let $\boldsymbol{\theta}$ be the vector of bar rotations associated with deformation into this deflected shape, and consider the one parameter family of deformations $\lambda\boldsymbol{\theta}$. The total potential energy of any member of

this family is

$$W = \frac{1}{2} K_G \lambda^2 \quad \text{with } K_G = \sum_{\text{bars}} (TL)\theta^2. \tag{2.12}$$

This is the key statement of this paper.

## 2.2. Graphical representation as weighted Maxwell–Minkowski

For the unimodal case considered, the stability analysis has been reduced to determining the sign of a single scalar quantity, the geometric stiffness $K_G = \sum (T/L)w^2 = \sum (TL)\theta^2$. Graphic statics possesses geometrical methods for representing all of the terms involved. The form diagram contains the information on bar lengths $L$ and the force diagram has the information on the tensions $T$. In two dimensions, the nodal displacements **a**, **b**, … etc. (and thence $w$) can be found using the incremental Airy procedure and can be unified with the forces in the Maxwell–Williot constructions of McRobie *et al.* [10]. The tension coefficients $\hat{t} = T/L$ manifest themselves as the aspect ratios of the intervening rectangles of side $T$ by $L$ that appear in the Maxwell–Minkowski construction. However, it is not clear how to make use of this aspect ratio in practice. Moreover, the concept becomes less evident for three-dimensional Rankine graphic statics. In the three-dimensional Rankine–Minkowski construction, the intervening objects between the components of the form and force diagrams are prisms of length $L$ with polygonal ends of oriented area $T$. There is thus no simple geometric interpretation of the tension coefficients in the three-dimensional Rankine construction.

The key to progress is thus to dispense with the $T/L$ tension coefficients in the $w$ formulation (equation (2.11)) and focus on the $TL$ load path terms in the $\theta$ expression of the total potential energy (equation (2.12)).

For two-dimensional trusses, this can be seen geometrically as the weighted sum of the areas $TL$ of the rectangles in the Maxwell–Minkowski diagram, with the weights being the square of the bar rotations. For three-dimensional trusses, the same weights are applied to the prismatic volumes of length $L$ and cross-sectional area $T$ in the Rankine–Minkowski diagram.

Note that the statement for two-dimensional trusses applies equally to both in-plane and out-of-plane stability. For the out-of-plane stability, a Rankine–Minkowski construction may be adopted which is simply the unit extrusion into the third dimension of the underlying two-dimensional Maxwell–Minkowski diagram. Everything may thus be represented diagrammatically in two dimensions, with no need to draw the unit extrusion explicitly.

The Maxwell–Minkowski rectangles and Rankine–Minkowski prism have clear geometric representation in terms of the unified form and force diagram, but the $\theta^2$ weights that arise from the mechanism motions need to be dealt with separately, and there are at least four methods of determining these. The first is via the standard matrix formulation of linear structural analysis: the nodal displacements of the mechanisms can be obtained by extracting a set of basis vectors spanning the null space of the compatibility matrix. The other three methods are more graphical. The first of these is to construct the Williot diagram, the basic graphical construction of small-displacement kinematics. Another approach which fits more closely with the overall graphic statics theme of this paper, and may be called the 'rotated incremental Airy' approach, is described in McRobie *et al.* [9]. Small incremental dual Airy stress functions over the force diagram generate small parallel motions of the form diagram, and 90° rotations of the nodal displacements lead to inextensional mechanisms. Finally, the 'sliding block' method of McRobie *et al.* [10] generates mechanisms by sliding cells of the force diagram along mutual edges, and is well suited to the identification of three-dimensional mechanisms. Here, we shall use the 'rotated incremental Airy' approach.

It follows that all the required terms in the summation (equation (2.12)) that determines the structural stability can be obtained by purely graphical means.

# 3. Examples: in-plane mechanisms of two-dimensional trusses

## 3.1. The pendulum

A simple pendulum consists of a bob of mass $m$ suspended by a weightless rigid rod hanging from a hinged support. Its natural frequency is $\omega = \sqrt{K/m}$ and yet according to elementary structural theory, a pendulum is a mechanism with zero stiffness, leading to the conclusion that the frequency is zero. Clearly, the gravitational force $mg$ acting on the mass provides tension-stiffening and the total

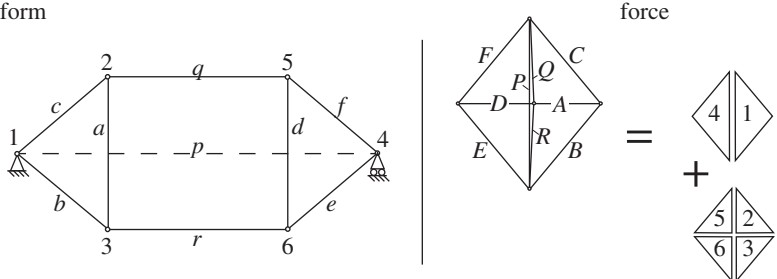

**Figure 3.** A simple tensegrity dome and its Maxwell reciprocal. The members of the outer hexagon *cqferb* are in tension while the two 'flying columns' *a* and *d* and the background horizontal member *p* are in compression. The force lines *Q* and *R* overlap the force line *P*, but have been drawn offset for clarity. This convention is used throughout the paper.

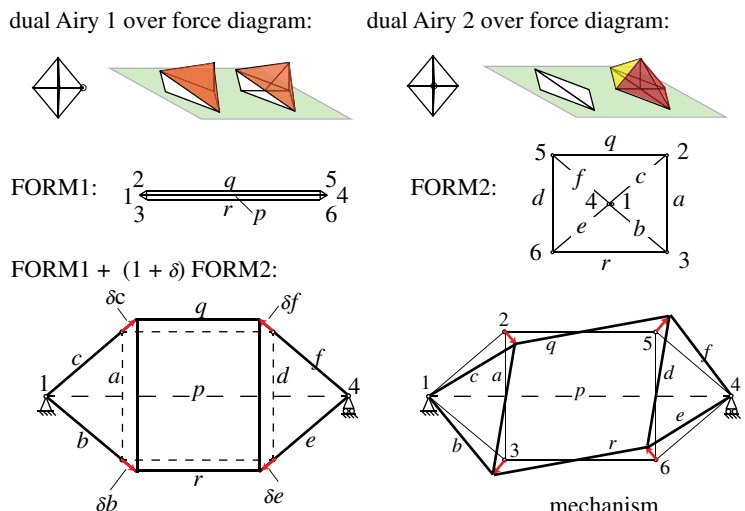

**Figure 4.** Creating the mechanism via graphical analysis.

potential energy summation of equation (2.12) associated with a small sideways motion $w$ involves only the single bar, giving $W = (1/2)(T/L)w^2$. The tension adds a lateral stiffness $K = T/L = mg/L$ and the frequency is thus $\omega = \sqrt{K/m} = \sqrt{g/L}$ as required.

## 3.2. Two-dimensional tensegrity dome

Figure 3 illustrates a simple two-dimensional tensegrity dome consisting of two outer triangles with an intervening quad. Two vertical compression struts act as 'flying columns', surrounded by the hexagonal tension loop which connects to the supports which are connected by the horizontal compression member $p$. The geometry is a manifestation of the famous Desargues configuration of three quads and two triangles that is central to projective geometry. The structure is a simple two-dimensional manifestation of the familiar three-dimensional 'suspendome' or Geiger dome [3,20].

In figure 4, the triangle 4 of the force diagram (shown in figure 3) is taken as the base plane, thereby fixing three of the five nodes. The two remaining nodes may be independently articulated out of plane to create two independent Airy stress functions over the force diagram. Each generates a corresponding form diagram, shown here as FORM1 and FORM2. Any linear combination of these two is also reciprocal to the force diagram. The original form diagram is one such linear combination. A further increment to the dual Airy stress function that created FORM2 keeps nodes 1 and 4 unchanged, but enlarges triangles *abc* and *def*. As previously, bars in the incremental diagram are parallel to their original directions, and a 90° rotation of the nodal increments leads to nodal increments which are perpendicular to the original bar directions, and thus correspond to an in-plane mechanism, as shown in figure 4.

The increments are shown in figure 5, with $\theta = w/L = \delta c/c$ for all members $a - f$ in the triangles. Factoring out the $(\delta c/c)^2$ term, their Maxwell–Minkowski rectangles thus have unit weight, being +1

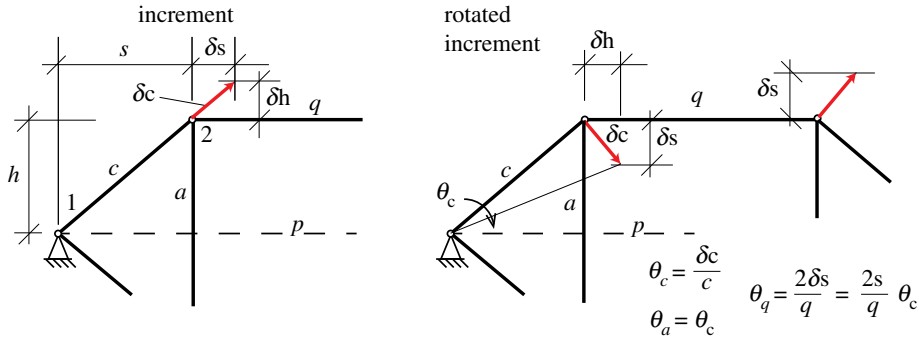

**Figure 5.** The bar rotations.

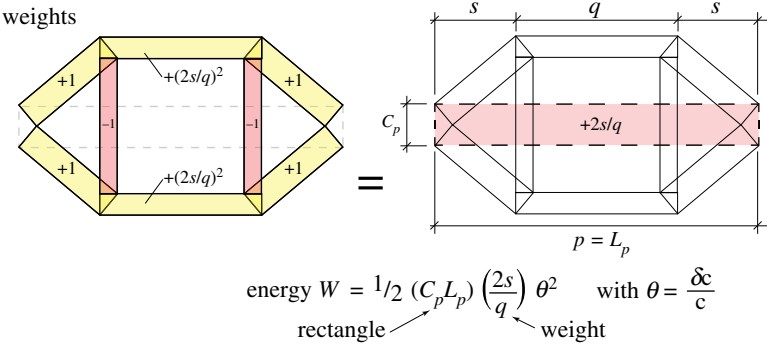

energy $W = \frac{1}{2}(C_pL_p)\left(\frac{2s}{q}\right)\theta^2$   with $\theta = \frac{\delta c}{c}$

rectangle          weight

**Figure 6.** The weights to be applied to the Maxwell–Minkowski rectangles.

for the sloping tension members and −1 for the flying compression struts. Via elementary geometry the horizontal tension ties $q$ and $r$ have $\theta = w/L = 2\delta s/q = (2s/q)(dc/c)$. The weights are shown in figure 6 and many have unit magnitude. It is known that if all weights had unit magnitude, appropriately signed for tension and compression, then the sum of the rectangular areas is zero. This is Maxwell's 1870 Load Path Theorem [2], and a proof follows readily from the rectangular areas of the Minkowski sum construction [12]. This result can simplify the equations. For the Geiger system the weighted sum is

$$W = \frac{1}{2}\left[4T_cL_c - 2C_aL_a + 2\left(\frac{2s}{q}\right)^2 T_qL_q\right]\frac{\delta c^2}{c^2} \tag{3.1}$$

and the Load Path Theorem gives

$$4T_cL_c - 2C_aL_a = C_pL_p - 2T_qL_q = C_p(p - q) \tag{3.2}$$

with the last step using $C_p = 2T_q$. Inserting this into the work equation gives

$$W = \frac{1}{2}C_p\left[p - q + \left(\frac{2s}{q}\right)^2 q\right]\frac{\delta c^2}{c^2}. \tag{3.3}$$

Since $p - q = 2s$ the term in square brackets simplifies to

$$2s + \frac{(2s)^2}{q} = \frac{2s}{q}(q + 2s) = L_p\frac{2s}{q}. \tag{3.4}$$

The work is thus

$$W = \frac{1}{2}C_pL_p\left(\frac{2s}{q}\right)\frac{\delta c^2}{c^2}, \tag{3.5}$$

which ascribes a weight $2s/q$ to the central rectangle. Since $2s/q$ is positive, it follows that the Geiger dome mechanism in this configuration is stabilized by the prestress. Somewhat remarkably, the stiffness and stability of the mechanism is most simply represented in terms of the weighted area of the Maxwell–Minkowski rectangle of the bar $p$ between the supports.

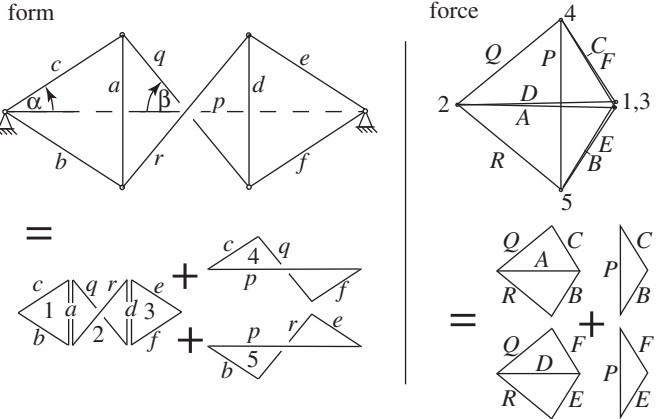

**Figure 7.** A variant on the tensegrity dome, together with its force reciprocal.

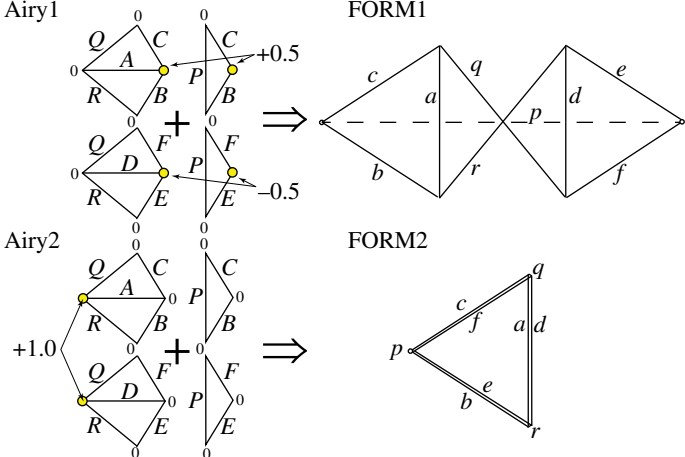

**Figure 8.** Independent dual Airy stress functions over the force diagram leading to a pair of independent form diagrams. The values of the dual Airy stress function are listed adjacent to the nodes of the force diagram.

## 3.3. The crossed tensegrity dome

Figure 7 shows a variation on the tensegrity dome of the previous example. The topology is identical but the geometry has been changed such that the central bars now cross. The force diagram is shown in figure 7, while figure 8 shows two dual Airy stress functions over the force diagram that lead to two independent form diagrams, FORM1 and FORM2. The FORM1 diagram has the required geometry for our structure, thus we increment the FORM2 diagram, rotate it by 90° and add to FORM1 to obtain the mechanism, as shown in figure 9.

In the mechanism, the inner crossed bars do not rotate, while all bars in the two outer triangles rotate by an angle of magnitude $\theta$. We can thus immediately invoke Maxwell's Load Path Theorem to eliminate the work done in the outer bars, and express the work solely in terms of the three central bars $p$, $q$ and $r$.

$$W = \frac{1}{2}\left[\sum_{a-f} TL\right]\theta^2 = -\frac{1}{2}\left[\sum_{p,q,r} TL\right]\theta^2. \tag{3.6}$$

Within the $TL$ summation over $p$, $q$ and $r$, the horizontal components of $q$ and $r$ cancel with that of $p$ over the central region, leaving the vertical components $T_v$ of $q$ and $r$ over height $2h$ and the horizontal component $C = 2T_h$ of $p$ over the two lengths $s$ outside the central region. Thus

$$W = \frac{1}{2}(-2T_v(2h) + (2T_h)(2s))\theta^2 \tag{3.7}$$

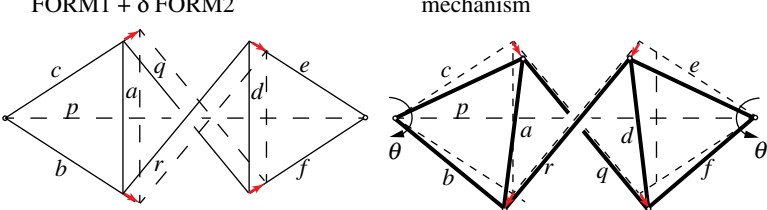

**Figure 9.** Constructing the mechanism by rotating an increment of FORM2.

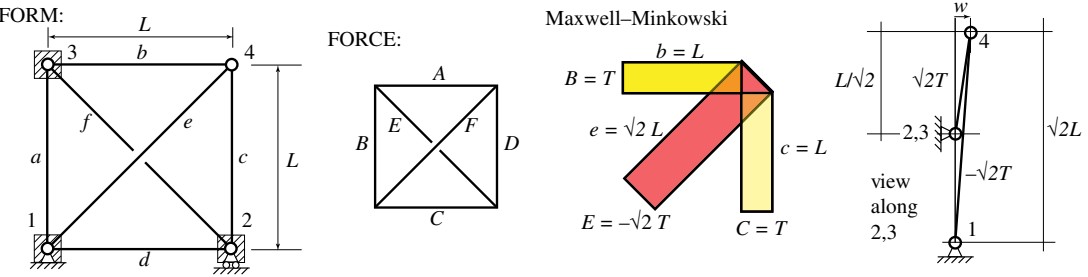

**Figure 10.** A self-stressed truss of K4 topology, together with its reciprocal, and the Maxwell–Minkowski diagram for relevant members. The projection obtained when viewing the buckled shape along the line between supports 2 and 3 is also shown.

which, after a little geometry, leads to the result

$$W = \frac{1}{2} K \Delta^2, \quad \text{where} \quad K = \frac{4T_c}{L_c} \frac{\cos(\alpha + \beta)}{\cos\alpha\cos\beta}, \tag{3.8}$$

where $\Delta$ is the downward displacement of the centre of either of the flying columns, and $T_c$ and $L_c$ are the tension and length of bar $c$.

For $\alpha$, $\beta$ each less than $90°$, the denominator is positive, thus stability is determined by the sign of $\cos(\alpha + \beta)$. Thus if the apex angle between the tension wires across the columns (between $c$ and $q$, say) is less than $90°$ then the system is unstable. This prediction was confirmed by simple experiments, as described in Millar *et al.* [15].

# 4. Out-of-plane mechanisms of two-dimensional trusses

Out-of-plane motions of two-dimensional trusses are essentially three-dimensional problems, and could thus be addressed using the methods of three-dimensional graphic statics such as Rankine reciprocal diagrams where bar forces are represented by the areas of polygons perpendicular to the bars. However, this is an unnecessary complication. Since both form and forces lie within the plane, they can be represented by two-dimensional graphic statics as usual, leading to the two-dimensional Maxwell–Minkowski diagram that combines both. The out-of-plane motions then lead to weightings of the rectangles of area $TL$ as before, there being nothing in the earlier derivation that required the nodal displacement vectors **a** and **b** (figure 2) to remain in the plane of the diagram.

## 4.1. Example: the cross-braced truss

This example, taken from Guest [8], checks the out-of-plane stability of a simple self-stressed two-dimensional truss. The truss has the full K4 graph, and thus corresponds topologically to Maxwell 1864 (fig. 1 in [1]), but with boundary conditions as per figure 10. Three nodes are constrained to lie in the two-dimensional plane, and there is a single mechanism in which node 4 moves out of plane, and involves bars $b$, $c$ and $e$ only. Bars in the outer quad carry a tension $T$ and the diagonals carry compressions of magnitude $\sqrt{2}T$.

When node 4 moves by $w$, the total potential energy is

$$W = \frac{1}{2} K w^2 \quad \text{with } K = \sum_{b,c,e} \frac{T_j}{L_j} = 2\frac{T}{L} + \frac{-\sqrt{2}T}{\sqrt{2}L} = \frac{T}{L}. \tag{4.1}$$

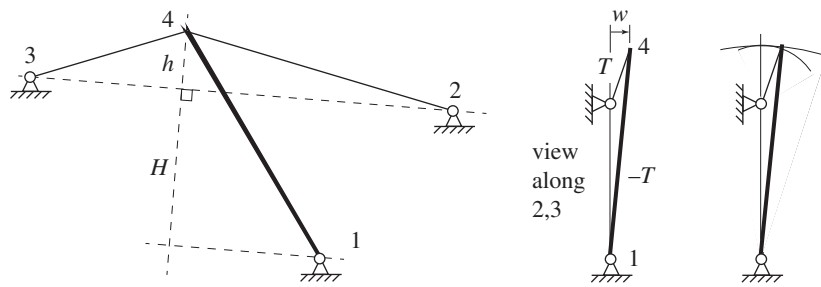

**Figure 11.** The clothes prop.

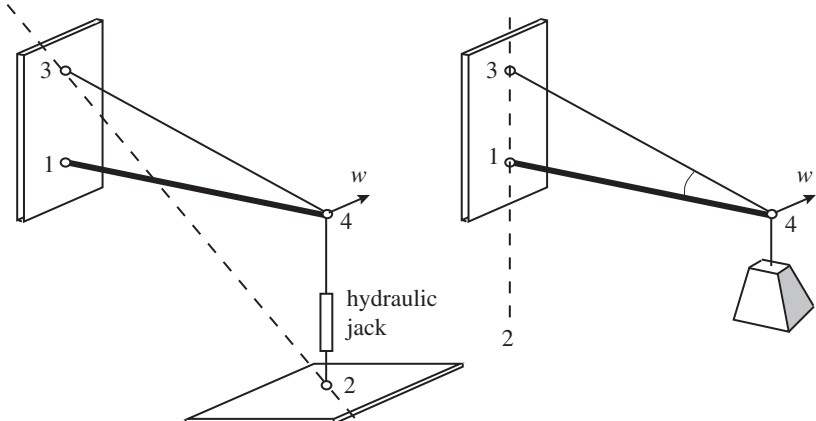

**Figure 12.** An undergraduate problem to design a simple cantilever.

Since the stiffness $K$ is positive, the state of self-stress stabilizes the out-of-plane motion of node 4. This conclusion can be obtained by inspection. The final drawing in figure 10 shows the projection looking along the line connecting supports 2 and 3. We may generally express $(T/L)w^2 = (TL)\theta^2 = Tw\theta$ for the contributions from the various components, and using the final expression, we note that the bar forces and end displacements are equal for the tension and compression load paths, but that the tension load path has the larger rotation $\theta$, thus the system is stable. The same analysis applies to the more familiar problem of a clothes prop propping a washing line as shown in figure 11.

Projecting onto the plane perpendicular to the line between the tension member supports, and using the $Tw\theta$ expression, it is again seen that the bar forces and end displacements are equal for the tension and compression load paths, but the tension members have the larger rotation $\theta$, thus the system is stable. This accords with the familiar explanation shown in figure 11 that inextensible members endeavour to rotate on circles of differing radii. If the prop is inextensible it will try to stretch the tension members out along the larger radius arc. These members will resist that, and this provides the stability. Stability would be lost if the prop base were to lie above the line 23 between the tension supports.

The same explanation applies to a structural design exercise undertaken by students early in the engineering degree at Cambridge University, requiring the design of a cantilever truss to be loaded at its tip by a hydraulic jack. A question arises as to whether the cantilever tip needs to be braced against out-of-plane deflection. Figure 12 shows the simplest solution, with a sloping tension bar and a horizontal compression bar. Again, looking along the line connecting the tension supports 2 and 3, it can be seen that the compression support 1 lies on the opposite side of the line 23 than the joint 4, and thus the system is stable. Were the jack to be replaced by a hanging dead load, the support 2 would effectively be at infinity and the compression support 1 would then lie on the line 23, and the system would be neutrally stable with respect to rotations around this vertical axis.

Figure 13 shows a potentially more economical solution to the design problem. When loaded with a suspended weight, there is neutral stability with respect to rotation about the vertical line of supports 13, meaning that lateral bracing is required to provide positive stability. By inspection, only one brace is required to provide the geometrically stable clothes prop configuration, either at the top or the bottom as shown.

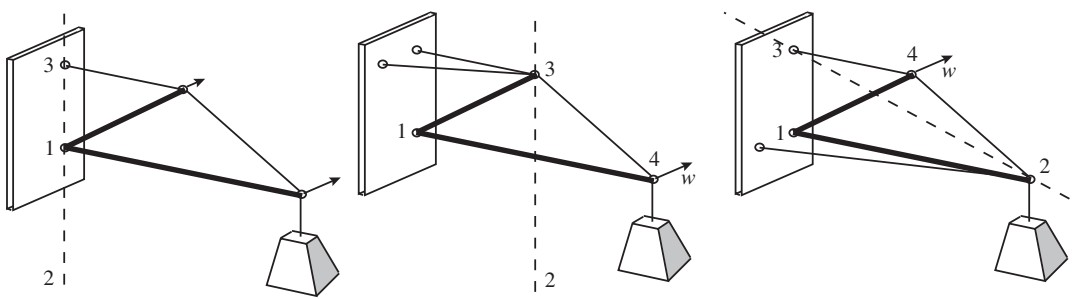

**Figure 13.** A more economical cantilever. Node numbering as per clothes prop.

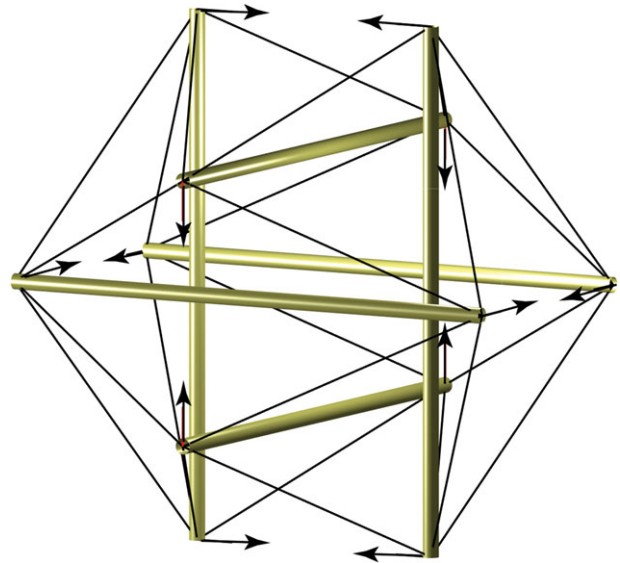

**Figure 14.** The Jessen icosahedral tensegrity is prestress stable because only the tension members rotate.

## 5. Fully three-dimensional mechanisms

Although all the previous examples were restricted to two-dimensional trusses the total potential energy summation over the bars was derived in three dimensions using figure 2 and is thus valid for the prestress stability of the mechanisms of three-dimensional trusses. The only difference concerns the graphical construction that uses the weighted sum of the Maxwell–Minkowski rectangles in two dimensions: this must now use the weighted sum of the Rankine–Minkowski prisms in three dimensions. That is, each $TL$ term is now represented as a volume given by the wedge product of the bar length vector with the polygonal area representing the tension in that bar.

We shall leave the treatment of three-dimensional trusses for later papers, as these will be more readily handled by the full multi-modal description that will be presented. However, we note in passing a few strong but simple results that follow almost trivially from the analysis here.

The first is that the Jessen icosahedral tensegrity is prestress stable (figure 14). It has a single mechanism in which each of the six compression members move laterally without rotation while the tension members in the surrounding web rotate. The work summation is thus necessarily positive which accords with the well-known observation that the mechanism is prestress stable. The second is that the cable-net roof designed by Buro Happold and Schlaich Bergermann Partner for the new Tottenham Hotspur Stadium (figure 15) is prestress stable [21]. The roof is essentially a three-dimensional version of the two-dimensional tensegrity dome of §3.2, with the central tensions being provided by two large tension rings that run around the tops and bottoms of the flying columns. The stiffness of the roof is provided completely by the geometric stiffness that results from the prestress.

Stability follows immediately by considering the triangular subframe consisting of a flying column (between the two tension rings) and the two radial tension 'spokes' that connect it to the surrounding compression ring. The geometric stiffness is most readily calculated by considering the subframe rotated to a vertical configuration and carrying weights which apply the radial components of tension from the tension rings, as per figure 15. This is equivalent to a pendulum of total weight $Mg = T$

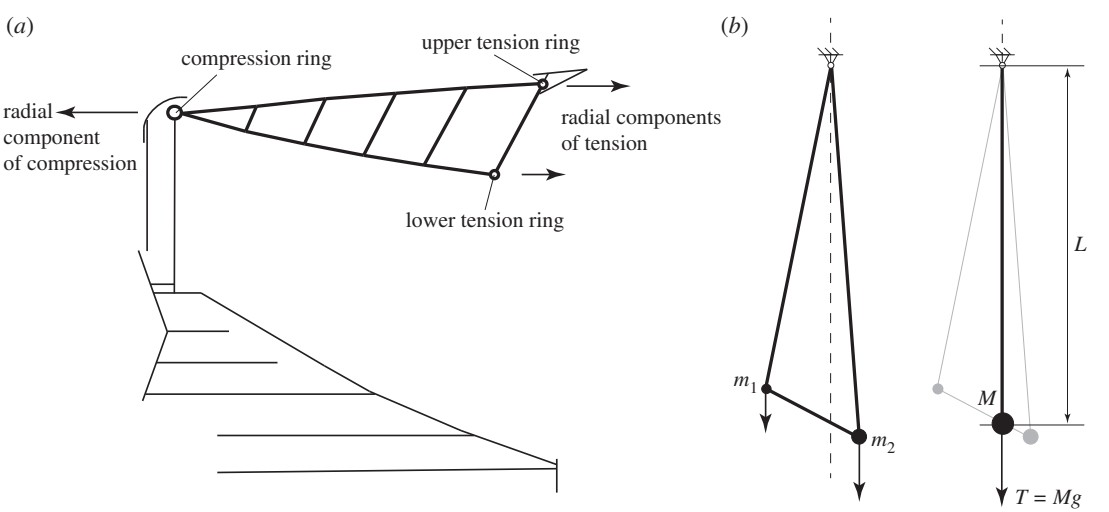

**Figure 15.** (*a*) The roof at the new Tottenham Hotspur Stadium is a cable-net tensegrity stabilized by prestress. (*b*) The geometric stiffness of a roof bay is revealed by rotating it to a vertical configuration and recognizing the equivalent pendulum.

(where $T$ is the overall tension at the support), with the pendulum bob located at the 'centre of mass' which must lie below the support. Elementary analysis shows the pendulum to have rotational stiffness $TL$ about any axis perpendicular to the pendulum axis, and zero stiffness against rotation about the pendulum axis itself, about which it can spin freely.

The stiffness of a mechanism of the whole roof will thus involve the sum of these non-negative stiffness contributions from the radial subframes, plus the additional geometric stiffness due to any rotations of members in the tension rings which—being tensile—necessarily give positive contributions to the sum. The structure is therefore prestress stable.

Within each subframe the roof panels are supported locally by compression struts which connect to the lower tension chord, and this arrangement is a more complex manifestation of the clothes prop in §4, and is necessarily stable. While it could be argued that graphic statics was not involved explicitly in the stadium roof analysis, it may also be argued that this is because the graphical analysis of the earlier examples of this paper have by this stage built up an intuitive fluency in identifying the key components that determine stability, allowing the stability of the complex structure to be distilled to that of the simple equivalent pendulum.

# 6. Product forces

During the wider process of putting the matrix analysis of structures into a rigorous framework of linear algebra, Pellegrino & Calladine [4,5,19] dealt with the possible prestress stiffening of mechanisms by introducing the notion of 'product forces' or 'geometric loads'. These are the nodal forces that must be applied to the structure in order to deform it into an infinitesimal mechanism, assuming that the pre-existing bar forces remain constant throughout the small motion.

We first provide the Pellegrino and Calladine matrix description, before giving the graphical statement of this, and showing how these relate to our earlier description. Elementary equilibrium requires applied nodal forces $\mathbf{f}$ to balance with bar tensions. Component-wise, that is

$$\sum_{\text{bars } j} T_j \cos \alpha_j = \sum_{\text{bars } j} \frac{x_i - x_k}{L_j} T_j = f_{ix}, \tag{6.1}$$

where $\alpha_j$ is the angle between bar $j$ and the $x$-axis, $f_{ix}$ is the $x$-component of the external force applied at node $i$, and $T_j$ is the tension in bar $j$ which connects to node $i$. Bar $j$ has length $L_j$ and connects node $i$ to node $k$, these having $x$-coordinates $x_i$ and $x_k$, respectively, in the equilibrium configuration.

If nodal forces $\mathbf{p}$ are applied to deform the structure from its original equilibrium position into a mechanism $\mathbf{m}$, then equilibrium at the new configuration requires

$$\sum_{\text{bars } j} \frac{(x_i + m_{ix}) - (x_k + m_{kx})}{L_j} T_j = f_{ix} + p_{ix}, \tag{6.2}$$

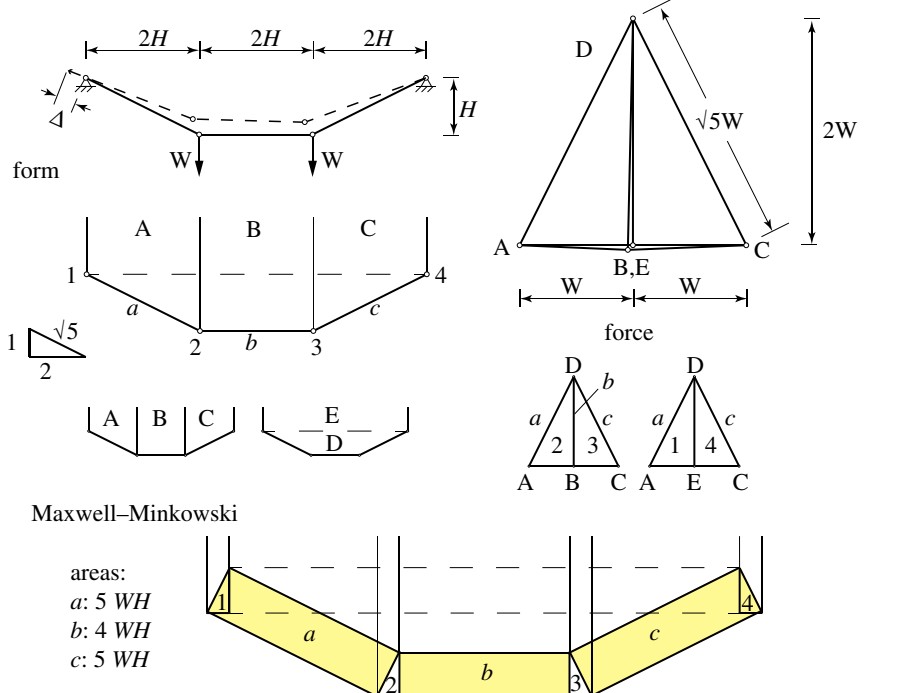

**Figure 16.** The hanging cable problem of Pellegrino [5]. The loaded structure hangs in a symmetric configuration and then the cable is wound in by a small distance $\Delta$ at the left-hand abutment. The form and force diagrams are shown for the original configuration, and their combination as a Maxwell–Minkowski diagram.

where $m_{ix}$, $m_{kx}$ are the $x$-components of the displacement at nodes $i$, $k$ of the unscaled mechanism $\mathbf{m}$. Subtracting the initial equilibrium leaves

$$\sum_{\text{bars } j} \frac{m_{ix} - m_{kx}}{L_j} T_j = p_{ix}, \tag{6.3}$$

where $m_{ix}$ is the $x$-displacement at node $i$ of the unscaled mechanism $\mathbf{m}$, node $i$ is the node where local equilibrium is being enforced, and bar $j$ connects node $i$ to node $k$. This defines the 'product forces' $\mathbf{p}$ in terms of the pre-existing bar tensions and the geometry of the structure and the mechanism. The aim here is to add graphical insight to this relationship.

We begin by following in detail the example given in Pellegrino [5] of a simple funicular holding two weights, and where the supporting inextensional cable is reeled in by a small amount at the left-hand abutment. Rather than solving the fully nonlinear constrained problem the aim is to estimate the final configuration using these linearized methods. Pellegrino provides the solution using matrix methods, but here we approach the problem graphically.

Figure 16 illustrates the problem, showing the form and force diagrams for the undeformed configuration. The form diagram involves a point at infinity where the lines of action of applied loads meet. The areas of the Maxwell–Minkowski rectangles are [5$WH$, 4$WH$, 5$WH$], respectively. If the displacements at nodes 2 and 3 are [$p_{2x}$, $p_{2y}$, $p_{3x}$, $p_{3y}$] = [1, −2, 1, 2]$\delta$ such that the bar rotations are [1, 2, 1]$\delta/H$, then the total potential energy is $(1/2)(5 + 16 + 5)(WH)(\delta/H)^2 = (1/2)(26W/H)\delta^2$, with prestress stiffness 26$W/H$.

We now apply compatibility to determine the effect of reeling in the cable from the left support. In this example, the cable is assumed to be inextensible. Compatible displacement diagrams can be readily constructed but they are not unique. Figure 17 illustrates those features of the displacement diagram that can be determined by first-order compatibility alone, namely:

— the abutment nodes 1 and 4 do not move and thus are represented by points at the origin of the displacement diagram;
— the point representing node 3 must lie on line 3 that rises at the 2 : 1 slope from the origin perpendicular to bar $c$;

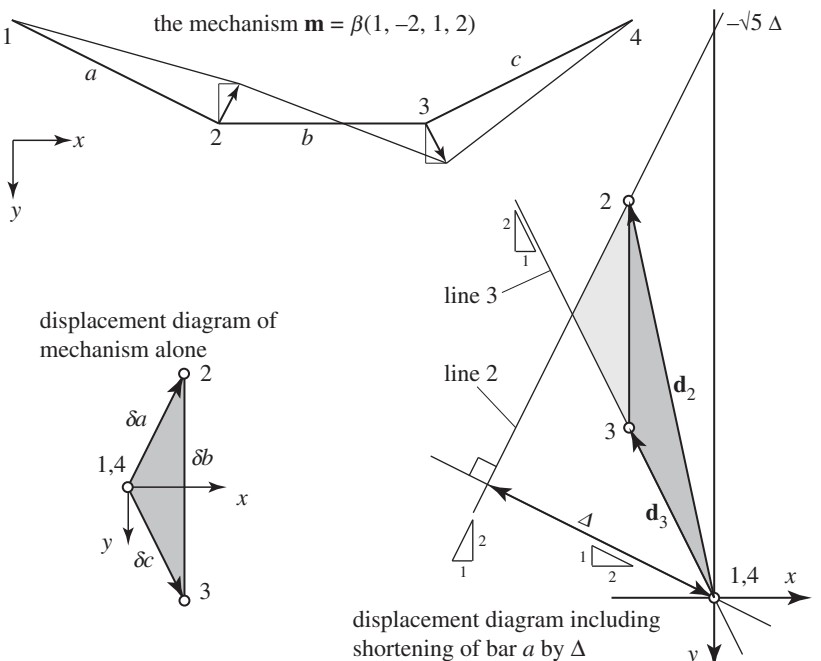

**Figure 17.** The inextensional mechanism and its displacement diagram are shown to the left. The lower right shows the overall displacement diagram including the bar $a$ shortening.

— the point representing node 2 must lie on line 2 that rises at the 2 : 1 slope and is offset from the origin by $\Delta$ in the bar $a$ direction;

— because nodes 2 and 3 are connected by the horizontal bar $b$ the points representing them must lie on a vertical line.

Figure 17 shows the displacement diagram for the mechanism alone, and how this fits into the overall displacement diagram including the bar $a$ shortening.

Any configuration satisfying these conditions is a compatible configuration. For example, in his matrix analysis, Pellegrino chooses the configuration $[p_{2x}, p_{2y}] = (0, -\sqrt{5}\Delta)$ and $[p_{3x}, p_{3y}] = (0, 0)$. These satisfy the above requirements, each lying on the $y$-axis of our displacement diagram. (For $\Delta = 10$ mm, $-\sqrt{5}\Delta = -22.36$ mm.)

However, addition of an arbitrary multiple of the inextensional mechanism $[p_{2x}, p_{2y}, p_{3x}, p_{3y}] = [1, -2, 1, 2]$ will also be compatible. The question is thus to find the appropriate combination which satisfies equilibrium.

The key is to identify what Pellegrino & Calladine [4,19] call the 'product forces'. These are the forces that need to be applied to the nodes to deform the structure into some particular mechanism, assuming that the bar tensions do not change.

These 'product forces' are illustrated in figure 18, focusing on bar $a$. First, all nodes are moved to their displaced positions. The product force at node 2 has a contribution associated with bar $a$. This is the additional external force $\mathbf{p}_{2a}$ that must be applied at node 2 to re-orient the old force so that it aligns with the new bar direction.

In the top left of figure 18, the tension $\mathbf{T}_a$ acting in bar $a$ on node 2 has first been rescaled to have the same length as the bar itself, such that its arrowhead is on node 1. In the next diagram, upper right, the structure has been flexed into the mechanism $\mathbf{m}$, and node 2 moves to the new location 2′. Node 1, being a support, does not move. The original force vector $\mathbf{T}_a$ no longer aligns with the new bar direction. In the diagram to the lower left, it is clear that a force $-\mathbf{p}_{2a}$ would reorient the bar tension to lie along the bar. In the final drawing, lower right, the product force $+\mathbf{p}_{2a}$ is thus applied externally at the displaced node 2′, and this is such that $\mathbf{p}_{2a} + \mathbf{T}'_a = \mathbf{T}_a$. That is, the product force plus the realigned bar tension equal the original tension.

It is clear from the geometry that the magnitude of the product force $p_{2a} = T_a \theta_a$. The distance which node 2 moves is the magnitude of the mechanism nodal displacement vector, which is $m_2 = L_a \theta_a$. The

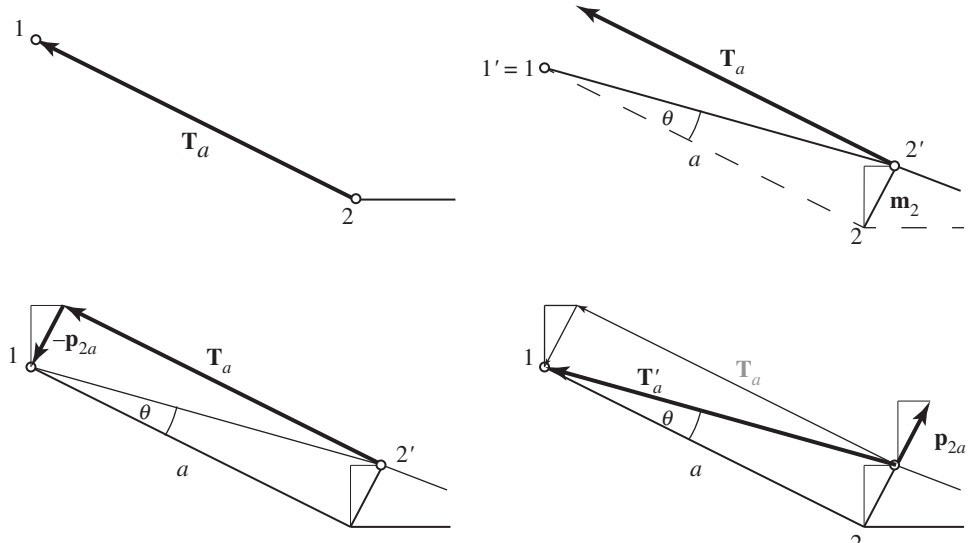

**Figure 18.** A geometrical construction for the contribution to the product force at node 2 from bar *a*.

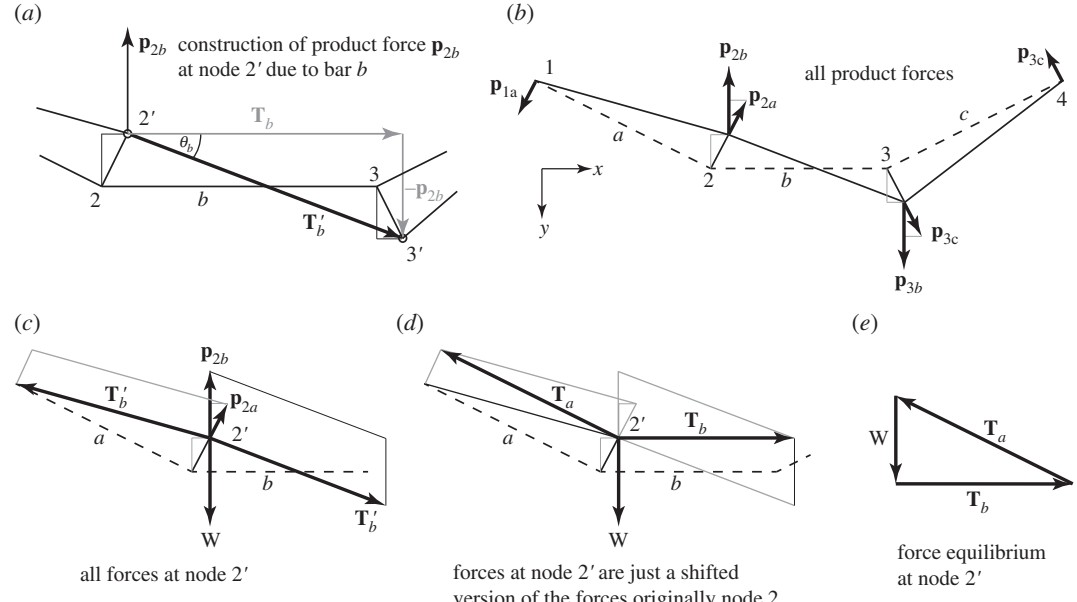

**Figure 19.** (*a*) The construction for the contribution of bar *b* at to the product force at node 2. (*b*) All product forces. (*c–e*) A graphical demonstration that the product forces restore the original equilibrium.

contribution to the total potential energy from the rotation of the force in this bar will equal half the applied force times the distance travelled, thus

$$\delta\Pi = \frac{1}{2}p_{2a}m_{2a} = \frac{1}{2}(T_a\theta_a)(L_a\theta_a) = \frac{1}{2}(T_aL_a)\theta_a^2. \tag{6.4}$$

It is often stated that the 'geometric stiffness' is given by the tension coefficient $T/L$. In the interpretation here, the product $TL$ can be seen as the *geometric rotational stiffness*, this being the stiffness against rotation of the bar due to the tension force it carries. The Maxwell–Minkowski diagram contains rectangles of area $TL$, and thus one interpretation of such a diagram is that it displays the geometric rotational stiffnesses of the stressed bars.

That is the key to the geometric understanding of the product forces. The product force at each node is the sum of the product forces due to each bar connecting at that node, and each individual product force is the product of the bar tension and the bar rotation as it undergoes the motion associated with the mechanism.

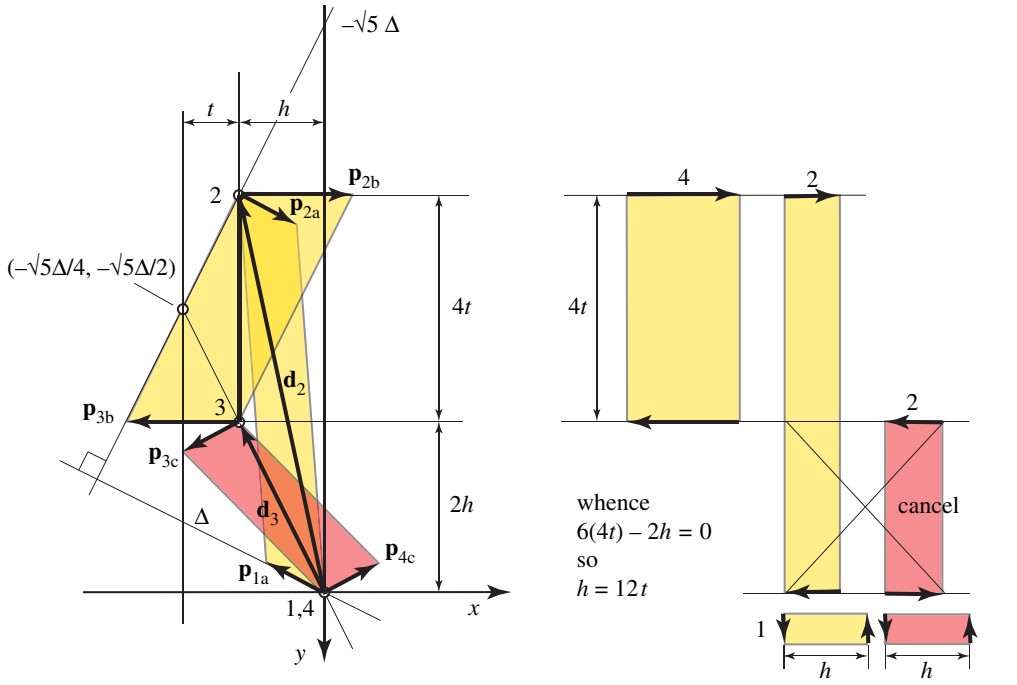

**Figure 20.** The work done by the product forces on the motion can be represented by oriented areas in a Maxwell–Minkowski form. At equilibrium, the product forces are orthogonal to the displacements, thus the total oriented area is zero. By elementary geometry, this requires the condition that $h = 12t$.

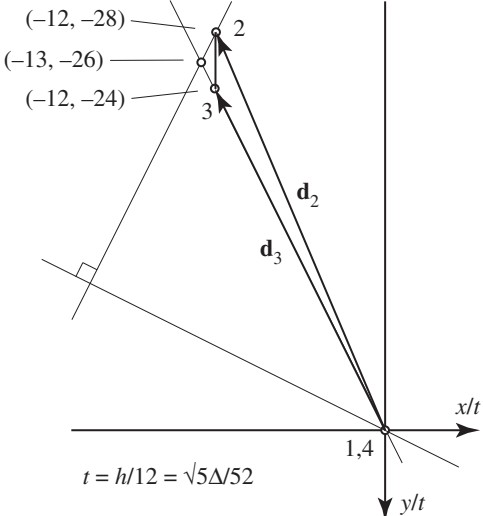

**Figure 21.** The solution.

To complete the analysis, we consider the other two bars. The geometry for bar $b$ is shown in figure 19$a$. Again, the product force required to align the original tension $\mathbf{T}_b$ with the new bar direction has the $T\theta$ form, namely $p_{2b} = T_b\theta_b$, and its direction is $\mathbf{m}_2 - \mathbf{m}_3$, the difference in the mechanism displacement vectors at the two ends of the bar.

The figure 19$c$–$e$ shows the set of all forces acting at the displaced node 2′ and how they are equivalent to the original equilibrium set at node 2. The mechanism has the form $[1, -2, 1, 2]$ and the corresponding product forces have the form $[1, -6, 1, 6]W/H$. It is interesting that the two shapes differ.

For the geometric construction, it is preferable to leave the product forces in terms of the separate contributions from each bar. For example, the total product force $\mathbf{p}_2 = [1, -6]W/H$ at node 2 will be kept as the separate contributions $\mathbf{p}_{2a} = [1, -2]W/H$ and $\mathbf{p}_{2b} = [0, -4]W/H$.

As explained in Pellegrino [5], at equilibrium, the product forces are orthogonal to the compatible displacements. Orthogonality may be expressed in terms of a dot product and in a manner akin to the

standard Maxwell–Minkowski diagram, the dot product of the nodal displacements with the product forces can be calculated graphically by first rotating the product forces by 90°. The dot product then becomes a cross product, whose magnitude is equal to the area of the parallelogram on the new unified diagram of mechanism displacement and product force. This is illustrated in figure 20.

At each node, the product forces are broken down into the contribution from each connecting bar, such that the contribution of each bar to the total summation can be represented by the area of the quadrilateral having the bar as one diagonal. In this case, the quadrilaterals are parallelograms. The sign of each oriented area is given by the direction of rotation of the product forces at either end of the bar.

Each parallelogram can be readily decomposed into constituent rectangles whose areas are readily computed. Orthogonality requires the total oriented area to be zero. Figure 20 shows that this requires the geometrical condition $h = 12t$, where $h$ and $t$ are defined in the diagram.

The solution so obtained is presented in figure 21. This agrees with the results of Pellegrino's matrix calculations and has been obtained by purely graphical methods. In summary, the displacement diagram of figure 17 was a qualitative schematic to represent the kinematic constraints, whereas the final diagram, figure 21, was obtained by graphical analysis of the product forces and is a quantitative representation of the actual displacements.

# 7. Summary and conclusion

This paper has shown how the geometric stiffness and prestress stability of pin-jointed trusses possessing a kinematic freedom may be determined graphically, without recourse to the matrix algebra that underpins earlier formulations. The paper is thus a further contribution to the recent renaissance of graphic statics, bringing stiffness and stability considerations within its compass. A key result was that the rectangular areas $TL$ in the Maxwell–Minkowski diagrams that unify the reciprocal form and force diagrams of a loaded truss give a powerful pictorial representation of the rotational geometric stiffness of the truss members that arise due to the forces in those members. Whether or not a given kinematic freedom is stabilized or destabilized by the presence of the internal forces is determined by the sign of a weighted sum of these stiffnesses.

The presentation here is restricted to the small displacements of two- or three-dimensional pin-jointed trusses with axially inextensional bars and with only a single kinematic freedom. The generalizations of this theory which describe the stiffness of trusses with numerous kinematic freedoms, with extensionable members and for large displacements are presented in companion papers [15,17,18], including the example application of the multimodal linearized theory to describe the stability of Robert Maillart's iconic design for the shed roof at Chiasso [16].

Data accessibility. This article has no additional data.
Authors' contributions. A.M. wrote the manuscript and created the figures, but all authors contributed equally to the development of the theory and to the reviewing of the manuscript.
Competing interests. The authors have no competing financial interests.
Funding. A.M. receives a salary from Cambridge University.

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
