## [Peer Review File · Royal Society Open Science]

Review History

RSOS-201970.R0 (Original submission)

Review form: Reviewer 1

Is the manuscript scientifically sound in its present form?

Yes

Are the interpretations and conclusions justified by the results?

Yes

Is the language acceptable?

Yes

Do you have any ethical concerns with this paper?

No

Have you any concerns about statistical analyses in this paper?

No

Recommendation?

Accept with minor revision (please list in comments)

Comments to the Author(s)

The authors are to be commended for a series of elegant graphical solutions to long-standing problems in structural mechanics. The problems have been solved previously elsewhere, but the authors confirm the results with new methodologies. This reviewer does not know of other similar work verifying stability through graphical solutions.

Major comments:

1) Given that the new methods are confirming previously known results, it would be helpful to better articulate exactly how the graphical methods are superior to matrix methods. Shorter computational time? More robust solutions? Easier to generalize? This reviewer would like to hear a clearer argument as to why graphical solutions are preferable to matrix solutions.

2) The authors mention companion papers and they cite several recent papers, particularly references [11, 13, 15, and 16], which they have published or submitted. It would be useful if they could clarify how the current paper differs from those previous papers.

Minor comments:

p 14 line 23: simple result[s] that follows almost trivially

It would be helpful to add a reference for the design of the Tottenham cable roof. A conference paper? A website?

Review form: Reviewer 2

Is the manuscript scientifically sound in its present form?

Yes

Are the interpretations and conclusions justified by the results?

Yes

Is the language acceptable?

Yes

Do you have any ethical concerns with this paper?

No

Have you any concerns about statistical analyses in this paper?

No

Recommendation?

Accept with minor revision (please list in comments)

Comments to the Author(s)

The paper provides a novel theoretical contribution by providing a new graphical solution to the stability of truss structures. More specifically, it provides a framework that enables a graphical depiction of the effective stiffness, and therefore stability. The paper focuses only on structures with "rigid elements", which allows the formulations to provide conceptual understanding of stability in an accessible manner. The authors appear to have extended this solution to non-rigid elements in another paper.

The paper is exceptionally well written, and the figures present difficult graphical concepts in a clear way. The following comments are all minor and are mostly suggestions to help further improve the clarity of the paper.

- Title: Suggest to remove “prestress” from the title. The method is not only applicable to prestressed structures, but simply stressed (or loaded) structures as well. However, the rigidity assumption is a big one, and this should be added to the title. Particularly in light of the fact that other recently submitted papers seem to focus on non-rigid structures, so the unique contribution of the paper is the rigid assumption.
- Intro: Suggest to clarify what is meant by “prestress” and clarify the structural determinacy for which the method is applicable.
- Page 3, line 58: State more specifically (here or later) what is meant by rotated 90 degrees. Also clarify whether this paragraph only refers to 2D, or 3D as well?
- Section 3a – Beautifully and very briefly demonstrates the concept.
- Figure 3 (& following): Suggest to simply add the bar “p” and make the right support a roller. The current presentation was at first confusing, since it wasn’t clear if p was there or not, or what the dashed line of p meant, and then “P” shows up in the force diagram. Alternatively, explicitly state that the solution which follows assumes that the dashed bar p exists, and the support forces are ignored.
- Figure 3 (right): Suggest to explain in the text that line P would overlap with lines Q and R, but these lines are drawn offset for clarity, and this convention is used throughout the paper.
- Page 7, line 56: maybe revise to “triangle 4 of the force diagram (in Fig. 3)”
- Figure 10 caption: suggest to change to “...when viewing the buckled shape...”.
- End of section 4: Suggested edit: “Fig. 13 shows a potentially more economical solution to the design problem”.
- Page 13, line 56: should “term” be “terms”.
- Page 14, line 24: should “follows” be “follow”
- Page 14, bottom: Suggest to add “between the two tension rings” to make the following: “consisting of a flying column (between the two tension rings) and two radial tension “spokes” that connect it to the surrounding compression ring”.
- Section 5: Tottenham roof: Agreed that the analysis nicely demonstrates stability, though the graphic statics contribution in this case is not particularly obvious. Does the graphic statics methodology shed any new light on this stability? Couldn’t the same thing be said using section 2a? Or the simple pendulum analogy?
- Figure 15 (right) = FYI. One of the gray arrows is still on the figure. The other is deleted.
- Figure 20 (right): The forces in this diagram should all be multiplied by W/H, correct?
- Figure 21: I see value in the proposed method to graphically display the effective rotational stiffness. This might be beneficial in design. However, what additional information does Figure 21 hold compared to 17? The benefit (to the structural designer, or otherwise) of the graphical product force method versus matrix methods or the displacement diagram in Figure 17 could be stated more explicitly at the end of this section.
- Final paragraph = is reference [15] one of the companion papers mentioned? If so, add reference to text.

Decision letter (RSOS-201970.R0)

Dear Mr McRobie

On behalf of the Editors, we are pleased to inform you that your Manuscript RSOS-201970 "Prestress Stability by Graphic Statics" has been accepted for publication in Royal Society Open Science subject to minor revision in accordance with the referees' reports. Please find the referees' comments along with any feedback from the Editors below my signature.

Please submit your revised manuscript and required files (see below) no later than 7 days from today's (ie 09-Mar-2021) date. Note: the ScholarOne system will 'lock' if submission of the revision is attempted 7 or more days after the deadline. If you do not think you will be able to meet this deadline please contact the editorial office immediately.

on behalf of Professor R. Kerry Rowe (Subject Editor)
openscience@royalsociety.org

Reviewer comments to Author:

Reviewer: 1

Comments to the Author(s)

The authors are to be commended for a series of elegant graphical solutions to long-standing problems in structural mechanics. The problems have been solved previously elsewhere, but the authors confirm the results with new methodologies. This reviewer does not know of other similar work verifying stability through graphical solutions.

Major comments:

- 1) Given that the new methods are confirming previously known results, it would be helpful to better articulate exactly how the graphical methods are superior to matrix methods. Shorter computational time? More robust solutions? Easier to generalize? This reviewer would like to hear a clearer argument as to why graphical solutions are preferable to matrix solutions.
- 2) The authors mention companion papers and they cite several recent papers, particularly references [11, 13, 15, and 16], which they have published or submitted. It would be useful if they could clarify how the current paper differs from those previous papers.

Minor comments:

p 14 line 23: simple result[s] that follows almost trivially

It would be helpful to add a reference for the design of the Tottenham cable roof. A conference paper? A website?

Reviewer: 2

Comments to the Author(s)

The paper provides a novel theoretical contribution by providing a new graphical solution to the stability of truss structures. More specifically, it provides a framework that enables a graphical depiction of the effective stiffness, and therefore stability. The paper focuses only on structures with “rigid elements”, which allows the formulations to provide conceptual understanding of stability in an accessible manner. The authors appear to have extended this solution to non-rigid elements in another paper.

The paper is exceptionally well written, and the figures present difficult graphical concepts in a clear way. The following comments are all minor and are mostly suggestions to help further improve the clarity of the paper.

- Title: Suggest to remove “prestress” from the title. The method is not only applicable to prestressed structures, but simply stressed (or loaded) structures as well. However, the rigidity assumption is a big one, and this should be added to the title. Particularly in light of the fact that other recently submitted papers seem to focus on non-rigid structures, so the unique contribution of the paper is the rigid assumption.
- Intro: Suggest to clarify what is meant by “prestress” and clarify the structural determinacy for which the method is applicable.
- Page 3, line 58: State more specifically (here or later) what is meant by rotated 90 degrees. Also clarify whether this paragraph only refers to 2D, or 3D as well?
- Section 3a – Beautifully and very briefly demonstrates the concept.
- Figure 3 (& following): Suggest to simply add the bar “p” and make the right support a roller. The current presentation was at first confusing, since it wasn’t clear if p was there or not, or what the dashed line of p meant, and then “P” shows up in the force diagram. Alternatively, explicitly state that the solution which follows assumes that the dashed bar p exists, and the support forces are ignored.
- Figure 3 (right): Suggest to explain in the text that line P would overlap with lines Q and R, but these lines are drawn offset for clarity, and this convention is used throughout the paper.
- Page 7, line 56: maybe revise to “triangle 4 of the force diagram (in Fig. 3)”
- Figure 10 caption: suggest to change to “...when viewing the buckled shape...”.
- End of section 4: Suggested edit: “Fig. 13 shows a potentially more economical solution to the design problem”.
- Page 13, line 56: should “term” be “terms”.
- Page 14, line 24: should “follows” be “follow”
- Page 14, bottom: Suggest to add “between the two tension rings” to make the following: “consisting of a flying column (between the two tension rings) and two radial tension “spokes” that connect it to the surrounding compression ring”.
- Section 5: Tottenham roof: Agreed that the analysis nicely demonstrates stability, though the graphic statics contribution in this case is not particularly obvious. Does the graphic statics methodology shed any new light on this stability? Couldn’t the same thing be said using section 2a? Or the simple pendulum analogy?
- Figure 15 (right) = FYI. One of the gray arrows is still on the figure. The other is deleted.
- Figure 20 (right): The forces in this diagram should all be multiplied by W/H, correct?
- Figure 21: I see value in the proposed method to graphically display the effective rotational stiffness. This might be beneficial in design. However, what additional information does Figure

21 hold compared to 17? The benefit (to the structural designer, or otherwise) of the graphical product force method versus matrix methods or the displacement diagram in Figure 17 could be stated more explicitly at the end of this section.

- Final paragraph = is reference [15] one of the companion papers mentioned? If so, add reference to text.

===PREPARING YOUR MANUSCRIPT===

===PREPARING YOUR REVISION IN SCHOLARONE===

Author's Response to Decision Letter for (RSOS-201970.R0)

See Appendix A.

Decision letter (RSOS-201970.R1)

Dear Mr McRobie,

It is a pleasure to accept your manuscript entitled "Stability of Trusses by Graphic Statics" in its current form for publication in Royal Society Open Science. The comments of the reviewer(s) who reviewed your manuscript are included at the foot of this letter.

on behalf of R. Kerry Rowe (Subject Editor)
openscience@royalsociety.org

Appendix A

Responses to Reviewers are shown in blue font.

Reviewer: 1 Comments to the Author(s)

The authors are to be commended for a series of elegant graphical solutions to long-standing problems in structural mechanics. The problems have been solved previously elsewhere, but the authors confirm the results with new methodologies. This reviewer does not know of other similar work verifying stability through graphical solutions.

Major comments:

1) Given that the new methods are confirming previously known results, it would be helpful to better articulate exactly how the graphical methods are superior to matrix methods. Shorter computational time? More robust solutions? Easier to generalize? This reviewer would like to hear a clearer argument as to why graphical solutions are preferable to matrix solutions.

2) The authors mention companion papers and they cite several recent papers, particularly references [11, 13, 15, and 16], which they have published or submitted. It would be useful if they could clarify how the current paper differs from those previous papers.

The authors are grateful to the Reviewer for the helpful comments.

We have added a paragraph at the end of the Introduction to indicate some of the advantages of graphical methods relative to matrix computation.

We have also added a paragraph to explain how this paper differs from our other papers.

Minor comments:

p 14 line 23: simple result[s] that follows almost trivially

Thanks. Corrected. ... simple results that follow...

It would be helpful to add a reference for the design of the Tottenham cable roof. A conference paper? A website?

Good point. A reference to a journal paper has been added.

Reviewer: 2

Comments to the Author(s)

The paper provides a novel theoretical contribution by providing a new graphical solution to the stability of truss structures. More specifically, it provides a framework that enables a graphical depiction of the effective stiffness, and therefore stability. The paper focuses only on structures with "rigid elements", which allows the formulations to provide conceptual understanding of stability in an accessible manner. The authors appear to have extended this solution to non-rigid elements in another paper. The paper is exceptionally well written, and the figures present difficult graphical concepts in a clear way. The following comments are all minor and are mostly suggestions to help further improve the clarity of the paper.

The authors are grateful to the Reviewer for the helpful comments.

- Title: Suggest to remove "prestress" from the title. The method is not only applicable to prestressed structures, but simply stressed (or loaded) structures as well. However, the rigidity assumption is a big one, and this should be added to the title. Particularly in light of the fact that other recently submitted papers seem to focus on non-rigid structures, so the unique contribution of the paper is the rigid assumption.

The title has been changed to remove the word "prestress". The title now reads "Stability of trusses by graphic statics". All three authors would prefer not to add the rigidity assumption to the title. The view was that this would make the title rather long and cumbersome. Most other papers on this subject that address stability by matrix methods, and the many allied papers on graphic statics also do not mention the axial rigidity of the bars in their titles.

- Intro: Suggest to clarify what is meant by "prestress" and clarify the structural determinacy for which the method is applicable.

These have now been clarified in the first paragraph.

- Page 3, line 58: State more specifically (here or later) what is meant by rotated 90 degrees. Also clarify whether this paragraph only refers to 2D, or 3D as well?

This has now been clarified.

- Section 3a – Beautifully and very briefly demonstrates the concept.
- Figure 3 (& following): Suggest to simply add the bar "p" and make the right support a roller. The current presentation was at first confusing, since it wasn't clear if p was there or not, or what the dashed line of p meant, and then "P" shows up in the force diagram. Alternatively, explicitly state that the solution which follows assumes that the dashed bar p exists, and the support forces are ignored.

The right-hand support has now been placed on rollers, such that the line p now corresponds to a bar.

- Figure 3 (right): Suggest to explain in the text that line P would overlap with lines Q and R, but these lines are drawn offset for clarity, and this convention is used throughout the paper.

This is a good point. A comment has now been added to the caption.

- Page 7, line 56: maybe revise to "triangle 4 of the force diagram (in Fig. 3)"

This has been changed and is now clearer.

- Figure 10 caption: suggest to change to "...when viewing the buckled shape...".

This has been changed and is now clearer.

- End of section 4: Suggested edit: "Fig. 13 shows a potentially more economical solution to the design problem".

This has been changed and is now clearer.

- Page 13, line 56: should "term" be "terms".

This has been corrected. Thank you.

- Page 14, line 24: should "follows" be "follow"

I apologise but I could not find this.

- Page 14, bottom: Suggest to add "between the two tension rings" to make the following: "consisting of a flying column (between the two tension rings) and two radial tension "spokes" that connect it to the surrounding compression ring".

Done. Thank you.

- Section 5: Tottenham roof: Agreed that the analysis nicely demonstrates stability, though the graphic statics contribution in this case is not particularly obvious. Does the graphic statics methodology shed any new light on this stability? Couldn't the same thing be said using section 2a? Or the simple pendulum analogy?

This is a good comment, and a sentence has been added to address this.

- Figure 15 (right) = FYI. One of the gray arrows is still on the figure. The other is deleted.

This has now been corrected. Also, the left hand figure has been made clearer.

- Figure 20 (right): The forces in this diagram should all be multiplied by W/H , correct?

This is an interesting point. Minkowski sums have the unusual property that units do not need to be consistent. Force vectors can be added to displacement vectors. Dimensional consistency could be obtained by normalising forces by a factor of H/W (or even $2H/W$, say) but there is no need: the condition that the areas of the intervening parallelograms sum to zero is independent of any such normalisation.

- Figure 21: I see value in the proposed method to graphically display the effective rotational stiffness. This might be beneficial in design. However, what additional information does Figure 21 hold compared to 17? The benefit (to the structural designer, or otherwise) of the graphical product force method versus matrix methods or the displacement diagram in Figure 17 could be stated more explicitly at the end of this section.

Text has now been added to make clearer the additional information contained in Fig 21 that is not in Fig 17, and how this information has been obtained graphically.

- Final paragraph = is reference [15] one of the companion papers mentioned? If so, add reference to text.

This and the other companion papers have now been referred to here, and in the Introduction.